# Infusing behavior science into large language models for activity coaching

**Narayan Hegde**[1][☺]*, **Madhurima Vardhan**[1][☺], **Deepak Nathani**[1], **Emily Rosenzweig**[2], **Cathy Speed**[3], **Alan Karthikesalingam**[3], **Martin Seneviratne**[3]

**1** Google Research, Bangalore, India, **2** Verily Life Sciences, San Francisco, United States of America, **3** Google Health, London, United Kingdom

☺ These authors contributed equally to this work.
* hegde@google.com

## Abstract

Large language models (LLMs) have shown promise for task-oriented dialogue across a range of domains. The use of LLMs in health and fitness coaching is under-explored. Behavior science frameworks such as COM-B, which conceptualizes behavior change in terms of capability (C), Opportunity (O) and Motivation (M), can be used to architect coaching interventions in a way that promotes sustained change. Here we aim to incorporate behavior science principles into an LLM using two knowledge infusion techniques: coach message priming (where exemplar coach responses are provided as context to the LLM), and dialogue re-ranking (where the COM-B category of the LLM output is matched to the inferred user need). Simulated conversations were conducted between the primed or unprimed LLM and a member of the research team, and then evaluated by 8 human raters. Ratings for the primed conversations were significantly higher in terms of empathy and actionability. The same raters also compared a single response generated by the unprimed, primed and re-ranked models, finding a significant uplift in actionability and empathy from the re-ranking technique. This is a proof of concept of how behavior science frameworks can be infused into automated conversational agents for a more principled coaching experience.

## Author summary

Sedentary lifestyle is strongly associated with long term adverse health outcomes. Digital apps provide new ways to motivate and promote physical activity at scale. Conversational assistants based on large language models (LLM) offer alternative to human coaches which is always available, economically viable and has access to growing findings in physical activity coaching science. For LLM coaches to be effective, they need to understand users need, context and strategies to change behaviors to resolve barriers and promote more activity. We propose novel techniques to infuse behavior science principles and understand context based on user queries to guide the LLM response to be appropriate to the context. We conduct blinded user studies to compare our work with native LLMs on multiple coaching attributes which promote sustained habits. Our techniques show better persuasion capability with empathy required for an effective digital coach. This work

**Data Availability Statement:** This study is predominantly based on simulated dialogue between an LLM and a member of the research team. Transcripts from coach-user conversations

in a previous study ((16)) informed study design, e.g. the design of the priming strategy and simulated dialogue. Raw data are not available to other researchers. Restricted access to the LaMDA model is available at https://bard.google.com. Priming text is included in Supplementary Materials. Study-specific code for BERT-based classifiers are released on Github at the following link, but no additional datasets are released: https://github.com/fitllm/classifiers.

**Funding:** This work was supported by Google LLC and/or a subsidiary thereof ('Google'). The funders had no role in study design, data collection and analysis, decision to publish, or preparation of the manuscript. All the authors received a salary from our funder (Google LLC).

**Competing interests:** The authors have declared that no competing interests exist.

provides qualitative instruments to guide further research in digital health coaching using LLMs and our methods can be applied to all types of dialogue based LLMs.

## Introduction

It is estimated that 81% of adolescents and 27% of adults do not achieve the levels of physical activity recommended by the World Health Organization; these levels are higher in developed countries (WHO) [1]. A sedentary lifestyle is strongly associated with long term adverse health outcomes, ranging from cardiovascular disease and diabetes to mental health problems and cognitive decline [2]. A 2022 WHO report found that Individual and group coaching to promote behavior change can be effective, but is not accessible to most, and effectiveness can be limited. Digital coaching tools have been highlighted as potential tools to address this gap [3].

Many digital health apps provide nudging to promote physical activity, as a low cost and scalable alternative to fitness coach. However, in an era of notification overload, there is also a risk of desensitization and alert fatigue if the nudge strategy is not well designed. Conversational agents can provide alternate strategy for physical activity coaching to provide more immersive experience and meaningful resolution of barriers. However, most traditional systems are limited in their degree of personalization and persuasiveness because they depend on rule-based nudge engines with static message content rather than adaptive conversational agents that can mimic realistic dialogue from a human coach, hence enhancing the capacity for personalization of the tool.

Large language models (LLMs), such as GPT-3 [4], PaLM [5], Gopher [6] and LaMDA [7], excel in natural language generation with greater expressivity and versatility compared to rule-based chatbots. LLMs Large language models (LLMs) are a type of artificial intelligence (AI) algorithm that can perform natural language processing (NLP) tasks. LLMs use deep learning techniques and massive datasets to understand, summarize, generate, and predict new content to perform human-like tasks such as generating and classifying text, answering questions in a conversational manner, translating text from one language to another and many more. To date, use of LLMs in the health and fitness space has been limited, however interest is growing rapidly following the release of LLMs tailored to biomedical tasks [8]. A major challenge in using LLMs in health care is how to ensure the model is personalized and adaptive while still remaining consistent with evidence-based practice and within safety guardrails [9]. Activity coaching relies on complex interpersonal dynamics where the coach builds rapport with the trainee, provides motivation, helps to overcome pre-existing patterns of behavior, etc.- which are not explicitly optimized in LLMs [10]. Knowledge infusion refers to the integration of established knowledge or practice into a model. In principle this is often achieved via finetuning on a task-specific dataset [11]. The disadvantage of finetuning in the coaching domain is that it requires coaching transcripts, which are difficult to obtain. Finetuning has also been shown to diminish the few-shot performance of a pretrained LLM with in-context prompts— i.e. over-specialization of the model [12]. Knowledge infusion is an active area of research and many other methods exist including customizing training objectives [13], reinforcement learning with human feedback [14, 15], in-context learning via prompt engineering or priming [16, 17] and many associated prompt design variants [18–21]. There have also been numerous strategies to ensemble knowledge infusion techniques, including post-hoc re-ranking or summarization of model outputs to further align the model with the task of interest [22, 23]. Customizing knowledge infusion strategies for the health care domain remains an area of active

research. Here we propose two simple in-context learning methods to infuse behavior science principles into LLMs without the requirement for finetuning or reinforcement learning.

Coaching in the context of physical activity ranges from delivering tailored products that serve elite athletes, to creating motivational tools that support inactive users to become fitter through progressive and personalised programs. Our LLM is designed to target the latter use case, to help users lead more active lifestyle using behavioral nudges and resolving barriers through conversations.

Behavioral science offers theoretical frameworks to help understand the factors influencing human behavior and design effective behavior change interventions for a given context. These framework combines elements of psychology, sociology, and anthropology to provide a scientific basis to interpret human behavior. COM-B is a well-known framework which conceptualizes behavior change along three axes: Capability (the psychological and physical skills to act); Opportunity (the physical and social conditions to act); and Motivation (the reflective and automatic mental processes that drive action) [24]. Behavioral science models can be useful to guide the design of automated nudging systems for habit formation [25].

The automated Physical Activity Coaching Engine (PACE) [26], is a chat-based nudge assistant tool that is based on an analogous behavior science framework called Fogg's Behavior Model (FBM), which focuses on 3 elements of behavior: motivation, ability, and a prompt. It was designed to boost (encourage) and sense (ask) the motivation, ability and propensity of users to walk and help them in achieving their step count targets, similar to a human coach. We demonstrated the feasibility, effectiveness and acceptability of PACE by directly comparing to human coaches in a Wizard-of-Oz deployment study with 33 participants over 21 days. We tracked coach-participant conversations, step counts and qualitative survey feedback. This rule-based automated nudging agent based on FBM had comparable outcomes to human coaches in terms of user step count and engagement. In this study, we extend findings of the PACE study by connecting the strengths of a behavioral science rule-based model with the conversational versatility of an LLM. The goal is to address the broader question of how behavior science principles might guide or constrain conversational LLMs. Specifically, we make use of priming and dialogue re-ranking. These are both lightweight techniques that do not require additional model retraining or finetuning. Overall, the key contributions contributions of this study are as follows:

1. Defining evaluation metrics for LLM conversations in the activity coaching domain

2. Introducing two different approaches to behavioral science knowledge infusion: coach phrase priming and dialogue re-ranking

3. Evaluating the benefit of knowledge infusion relative to an unprimed LLM using quantitative and qualitative approaches

## Related work

Numerous smartphone nudging tools have been designed to promote physical activity [27, 28]. These interventions are low-cost and highly scalable relative to human fitness coaches, with promising early evidence [29–32]. The general findings suggest that self-monitoring and goal-targeting can enable users to better integrate physical activity and guide them in adopting healthier lifestyle [33–35]. The commonly used intervention strategy by digital fitness apps has been push notifications comprising of exercise reminders to prompt users to exercise [36, 37]. Furthermore, researchers have designed features for app-based prompts to be user adaptive, either with respect to timing or frequency [38–41]. The inclusion of personalization in fitness

apps has shown promising results, such as improving trends of user physical activity with a passive smartphone-based intervention without the need of external human coaching [36–39, 41, 42]. Such developments in fitness apps are incredibly pivotal, given the current pandemic scenario and the shortage of trained fitness coach practitioners [40, 43, 44]. However, in an era of notification overload, there is also a risk of desensitization and alert fatigue if the nudge strategy is not well designed [45, 46].

Automated conversational agents offer an opportunity to create interactive dialogue, with widespread applications in e-commerce, home automation and healthcare [47, 48]. Health and Fitness coaching is emerging as a promising use case for these conversational agents [26, 49–51]. AI and rule based conversational agents have been studied to assist in selfcare, mental and physical health care management ecosystems and promoting physical activity [52–57]. Common challenges facing interventions were repetitive program content, high attrition, technical issues, and safety and privacy concerns. However, most traditional systems are limited in their degree of personalization and persuasiveness because they depend on rule-based nudge engines with static message content rather than adaptive conversational agents that can mimic realistic dialogue from a human coach [58], hence enhancing the capacity for personalization of the tool.

Our conversational fitness agent is based on publicly available Large Language Model to provide more naturalistic conversation on fitness challenges and cover wider range of topics. To our knowledge, this is the first adaptation and evaluation of LLMs for physical activity coaching by inducing behavior science principles. Our method is easy to replicate on new LLM models which are being trained on larger and more diverse datasets to finetune for physical activity coaching usecase.

## Methods

The following sections outline the datasets, language modeling techniques and evaluation methods used.

### Data

The previous PACE study dataset was re-purposed for this analysis [26]. Specifically, this dataset was used to construct the example coaching phrases used in the behavior science priming, create training data for finetuning Bidirectional Encoder Representations from Transformers (BERT) [59] user and coach statement classifiers and to select the user queries (initial user responses) in simulated conversations for evaluation. This dataset consists of dialogue transcripts between fitness coaches and subjects, generated from real coaching interactions across various physical activity habit formation related issues. The consenting subjects were randomized to either have fitness conversation with human coaches, or chatbot assistant that suggested example responses based on FBM behavior science using a rule-based engine. The chatbot was interfaced to participants using WoZ(Wizard of Oz) method, which is a common approach used for testing human-robot interaction allowing us to substitute natural language understanding and generation tasks by keeping a human in the loop. The dataset included 520 + conversations from 33 participants over 21 days. A total of 6 independent annotators labeled these conversations as one of Motivation, Capability and Opportunity. We determine the user state on three fronts: motivation, capability and opportunity each of which being high, low, and unknown. To this end, we rely on the conversation engagement patterns and use the information about the previous day step count. The coach actions were first evaluated whether corresponding to sense or boost. Boost was further annotated on the same three criteria as user state: motivation, ability and propensity. Both user and coach statements where separately

annotated with presence or absence of each of these three themes. Data collection and annotation protocol is described in detail in [26].

## Language models

The Language Models for Dialog Applications (LaMDA) pretrained LLM was used as the primary architecture [7], with no further finetuning. But unlike most other language models of past, LaMDA is trained on dialogue and conversation datasets. During its training, it picked up on several of the nuances that distinguish open-ended conversation from other forms of language. The auto-completion is tuned for sensibleness and specificity. LaMDA is a decoder-only transformer architecture with 64 layers, used here in its 137 billion parameter configuration. We used the following LaMDA hyperparameters: temperature 0.9; maximum token length 1024, top k (controls sampling diversity) 40. LaMDA has an option to provide context alongside the LLM prompt—this was how the coach phrase priming was conducted. LaMDA also provides top-k outputs, which were used in the re-ranking (see below).

## Coach phrase priming

Priming (also called Prompt engineering) is the process of creating a snippet of text called prompts for LLMs to generate a desired output. Prompts can include instructions, few example input and outputs, questions, or any other type of input, depending on the intended use of the model. Coach phrase priming was performed by inputting 30 example coach nudges as context to the LLM prior to the prompt. This priming anchors the conversation to look like user-coach interaction by giving examples of common scenarios encountered by coaches. The 30 nudges were selected from the data in the PACE study—specifically the 10 most common coach responses in each of the three behavior science categories of interest: C/O/M. Details regarding the coach phrase selection and priming method are described in S1 Text and S1 Table. For example, the Capability category included activity planning and barrier conversations; and Opportunity included social engagement conversations and activity planning; and Motivation included congratulations and positive affirmation; [24]. The order of the 30 nudges was randomized. The priming prompts are shown in Table 1.

## Simulated dialogue

The following LLM configurations were compared via simulated conversations with a single member of the research team:

1. Unprimed (only user query is given as prompt)

2. Coach-primed (30 example nudges provided as LLM context along with user query as prompt)

All conversations begin with the trigger prompt: *Hey John, It's time for your morning walk.* The subsequent user responses were sampled from a set of 9 user statements, with 3 each designed to evoke a low Motivation, low Capability and low Opportunity (fitness related user queries are included in the S2 Table). An example user statement with low opportunity was: *I am super busy with work today. I have chores to do in the morning and work meetings after that.*.

This culminated in a total of 18 transcripts: 9 each for the unprimed and primed LLMs. The conversations were continued with dialogue between the LLM and the human interlocutor (researcher). The conversations were terminated at a natural breakpoint at the discretion of the researcher. Any follow up questions to the LLM response were added appropriately to

**Table 1. LLM prompts used in coach phrase priming.**

| Behavior Science Priming |
| --- |
| The following is a conversation with an AI Health Coach. <br> The coach tries to motivate the users when the user lacks motivation, can resolve barriers. <br> Here are some examples of how a coach can help users: <br><br> "I know you probably have a busy schedule. I still think <br> you can manage and hit your goal of daily step count." <br> "Looks like you are having a busy day. I would recommend <br> setting up gentle reminders daily of your goal to have them <br> as part of each day. Hope that can help you be all set for having an exercise routine!" <br><br> "You know, building a new habit is really really hard. But it doesn't have to be that way:) <br> Starting with a little stroll outside for some fresh air cannot be bad idea as long as the <br> weather is right. So why not head out today for a few minutes, and come in. What do you think?:)" <br><br> "You must keep that fire burning, your excitement and confidence for maintaining <br> a healthy lifestyle will take you far with healthy habit formation. <br> I believe a daily stroll with be no problem for you at all:)" <br><br> "So do you reckon you'll manage your walk today?" <br><br> "It is nice and bright outside today. What is your plan for the day, why not start walking today?" <br><br> "The question you can ask yourself is that do you feel walking can help you?" <br><br> "You knew starting a healthy habit can be hard, but it's a life changing <br> experience of rebuilding your identity as someone who exercises:) <br> If you're not feeling up for a long walk today, perhaps we can aim for a shorter one?:)" <br><br> "You know walking can be especially enjoyable as it allows you to put <br> on your favourite playlist and podcast. So, what do feel like listening to today?" <br><br> . . . |
| **Coach prompt: Hey John, It's time for your morning walk**. |

continue the conversation on the original topic until a logical end was reached. Additional example transcripts are contained in the S2 Fig.

## Constraining LLM responses using a COM-B classifier

In order to further constrain or guide the LLM to provide nudges based on COM-B principles, we trained two classifiers to assess C/O/M levels:

1. User statement classifier: Given a user statement sentence, the user-query classifier assigns a high vs low value for each of the capability, opportunity and motivation(COM) dimensions (multi-label classification).

2. Coach statement classifier: Given a shortlist of 15 top LLM outputs, the coach response classifier maps each response to either C, O or M (multi-class classification).

The classifiers were designed as follows. The input string (could be multiple sentences) was embedded using a BERT-base model with the final layer finetuned over either a multi-label head (user statement classifier) or 3 separate C/O/M heads (coach statement classifier). Models were optimised with a cross-entropy loss. Separate user and coach classifiers were trained using samples of 432 user statements and 531 coach statements from the PACE study, manually annotated with C/O/M status as detailed in S4 Text. These datasets were split 70:10:20 across train, validation and test splits. Weights were not shared between the user and coach models.

## Simulated dialogue with re-ranking

The simulated conversation experiment was repeated with the primed LaMDA model, using the above classifiers to align the coach response to the inferred user need. For the 9 coach-primed LaMDA transcripts above, a single user statement was manually selected as the most representative of the user's behavioral need.

LLMs are trained to generate the next word and sentence based on given text context. While generating the next token in the input sequence, the model comes up with a probability distribution for all words. The temperature parameter adjusts the shape of this distribution, leading to more diversity in the generated text. Top-k tells the model that it has to keep the top k highest probability tokens, from which the next token is selected at random. LLMs generate many sentences for a given user query and one of them is selected as response. This strategy works well for generic conversation. But the response may not adhere to behavioral model which is designed to be context sensitive and understand user query in relation to fitness barriers. The re-ranking method orders the top 15 LLMs responses based on COM-B framework for the context defined by user query. The top of the ordered list matches the response needed to address the user barrier for pursuing physical activity. For example, person seeking ways to make walking fun should receive ideas like temptation bundling or walking with friends and not foot-in-the-door or perceived benefits as suggestions.

The selected text was input into the user statement classifier to identify the C/O/M need. The same user text was input into the coach-primed LaMDA model to generate the top 15 candidate responses. These 15 responses were then separately run through the coach statement classifier to generate a likelihood score across each C/O/M category. The coach action was re-ranked based on the user's inferred C/O/M need based on the rules in Table 2 (i.e. the statement with the highest C/O/M likelihood score in the desired coach action was chosen). More examples of coach response and user statement classifier are provided in the S3 Text. After re-ranking LLMs responses, the top-1 result is given as response to user query. An example conversation post dialogue re-ranking is shown in Fig 1. The user query to LLM response flowchart is shown in S2 Fig.

In addition, we conducted an 'oracle' experiment where the user response was manually categorized into C/O/M need and the corresponding coach-primed output was chosen.

Two manual review exercises were then conducted:

1. Comparing the coach-primed output to the classifier re-ranked output; and

2. Comparing coach-primed with the oracle re-ranked output. Note that in both these review exercise, only a single coach response was being adjudicated rather than an entire conversation as previous.

## Evaluation attributes

An evaluation framework was defined based on four key attributes of an LLM-based fitness coach: actionability, realism, motivation and empathy. Table 3 shows how these attributes

**Table 2. Decision matrix to select nudge theme based on C/O/M values derived from the user statement classifier.**

| Capability | Opportunity | Motivation | COM-B Action |
|---|---|---|---|
| Low | High/Low | High/Low | Boost Capability |
| High | Low | High/Low | Boost Opportunity |
| High | High | High/Low | Boost Motivation |

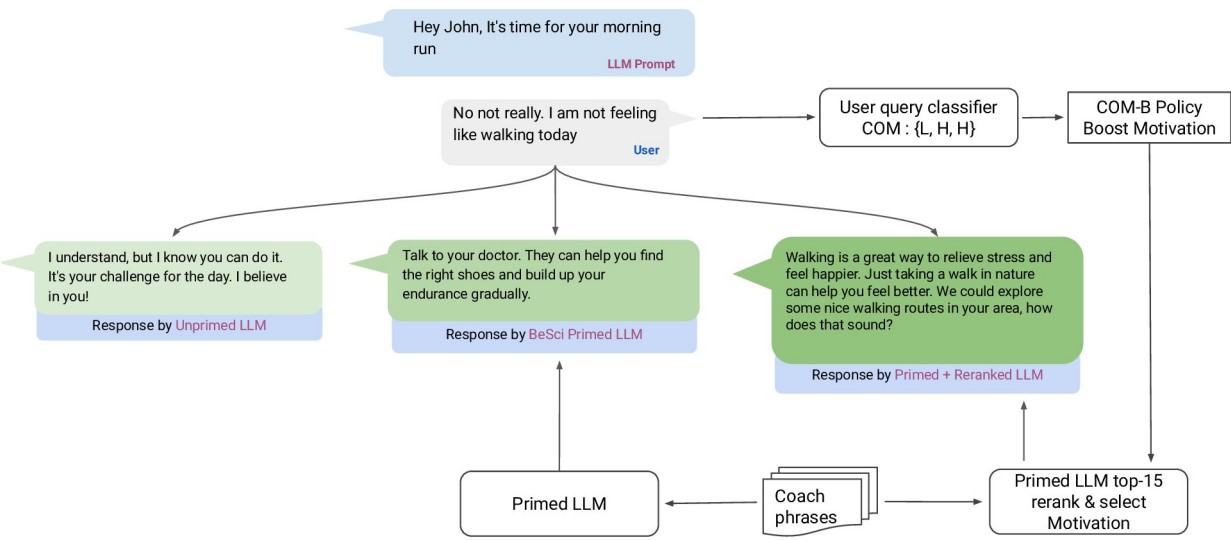

**Fig 1. Comparison of example conversations with unprimed, coach-primed and primed+reranked LLMs.**

align with published evaluation frameworks for coaches [60] and for LLMs [7, 61]. Eight independent reviewers rated the transcripts on the four attributes (actionability, realism, empathy, motivation) and to rate the transcripts for overall quality. Reviewers were blinded to the manner of LLM priming (naive vs coach) and Re-Ranked LLM variations. Raters assessed each pair of naive-primed and coach-primed transcripts on the four attributes using a Likert Scale ranging from 1–5. The specific rating prompts and likert scale labels are included in S2 Table and S2 Text. S1 Fig shows the conversation transcript tool built to rate the coach LLM and user conversation session to test the efficacy of the proposed LLM enhancements.

We also evaluated the conversations based on a set of additional quantitative conversational quality metrics, including average length of reply, number of conversational turns, user sentiment at conversation end, presence of questions in the coach dialogue, and use of coaching-specific words ('goal', 'health', 'routine', 'recover', 'challenge', 'workout', 'training', 'rest').

## Statistical Analysis Plan

To determine whether the evaluation attribute ratings for primed and unprimed models differed from each other, we ran a series of linear mixed models, one for each of the four attribute ratings, and one for the global quality rating. These included a fixed intercept $\beta_0$, fixed effect for primed vs unprimed condition $\beta_1$, and random intercepts for rater $\mu_0$ and conversation prompt $\mu_1$ to account for the non-independence of the observations. The equation for each of these models is: $Y_{ij} = \beta_0 + \beta_1 * X_{ij} + \mu_0 i + \mu_1 j + \epsilon_{ij}$

**Table 3. Evaluation attributes cross-referenced with established attributes of coaches and LLMs.**

| Evaluation attributes | Coach attributes | LLM attributes |
|---|---|---|
| Actionability | Professional competence | Informativeness |
| Realism | Context sensitivity | Sensibleness & safety |
| Motivation | BeSci interventions | Interestingness |
| Empathy | Social-emotional competences | Groundedness |

To determine whether the primed and unprimed models differed from each other on the five quantitative conversational quality metrics, we conducted a paired-sample t-test for each metric. For example, we compared the average length of the LLM reply for the paired primed and unprimed responses to each of the 9 conversation prompts.

## Results

The evaluation attribute ratings of blinded reviewers were overall more favorable for the coach-primed versus the unprimed LLMs. Specifically, conversations produced by the coach-primed model were rated as significantly better than conversations produced by the unprimed model on overall quality, providing actionable suggestions, and using realistic language.

The ratings for the classifier re-ranked versus unprimed output were less conclusive, but this may be because those ratings were based on a single statement response from the model rather than a full back-and-forth dialogue. Based on Likert scale responses, the re-ranked answers were rated as more actionable, realistic & empathetic; The better performance is attributed to appropriate response matching for user query using classifier based re-ranking.

Examining the quantitative conversational quality metric data Table 4: the number of turns of dialogue, number of questions asked of the user, and number of coaching-related words were each significantly higher for the coach-primed versus unprimed conversations. Across both architectures, priming was associated with a significant boost in the rate of conversations ending in a positive user sentiment, the rate of question-asking by the coach LLM, and the use of coaching-related vocabulary.

To determine whether ratings for the primed and unprimed models differed from each other, we ran a series of linear mixed model analyses. These included a fixed effect for primed vs unprimed, and random effects for rater and prompt to account for non-independence of the observations. Regarding message content, the ratings of blinded reviewers were overall more favorable for the coach-primed LLMs as shown in the Table 5. Specifically, the coach-primed model was rated as significantly higher in terms of quality, providing actionable suggestions, and using realistic language.

**Table 4. Quantitative conversational quality metrics for unprimed versus coach-primed LLM conversations.**

| Metric | Unprimed | Coach-primed | p value |
|---|---|---|---|
| Average length of LLM reply (# words ± S.D.) | 25.7 ± 6.5 | 23.7 ± 7.1 | 0.42 |
| Turns of conversation by user/LLM (# turns ± S.D.) | 3.1 ± 0.3 | 3.7 ± 0.7 | 0.17 |
| Conversations ending with positive user sentiment (%) | 30 | 60 | **0.01** |
| Conversations containing a question asked by LLM (%) | 0 | 30 | **0.04** |
| Conversations containing coaching-specific words used by LLM (%) | 40 | 80 | 0.08 |

**Table 5. Evaluation attribute ratings for unprimed versus coach-primed LLM conversations.**

| Survey question (1, strong disagree 5, strong agree) | Unprimed | Coach Primed | p value | Beta | Cohen-D |
|---|---|---|---|---|---|
| *Which conversation provides a better overall coaching experience (%, remainder unsure)* | 21 | 72 | - | - | - |
| *The quality of the coaching experience is high* | 3.28 ± 0.88 | 3.78 ± 1.0 | **<0.001** | 0.51 | 0.37 |
| *The coach provides concrete fitness strategies that are actionable to the user* | 3.43 ± 0.78 | 4.05 ± 0.84 | **<0.001** | 0.62 | 0.45 |
| *The coach provides motivation or encouragement to the user* | 3.5 ± 1.1 | 3.78 ± 0.96 | 0.10 | 0.28 | 0.19 |
| *The coach is empathetic toward the user's needs and challenges* | 3.36 ± 1.04 | 3.64 ± 1.09 | 0.09 | 0.28 | 0.19 |
| *The language used by the coach is realistic and appropriate for the setting* | 3.53 ± 1.09 | 3.91 ± 1.01 | **0.02** | 0.38 | 0.26 |

**Table 6. Class balance and model performance on C/O/M classification for user statements.**

| Category | Class balance (high:low) | | Classifier performance (ROC-AUC) |
|---|---|---|---|
| | Train | Test | |
| Motivation | 220:40 | 68:16 | 0.86 |
| Capability | 158:66 | 51:40 | 0.77 |
| Opportunity | 112:34 | 52:13 | 0.83 |

Tables 6 and 7 show the performance of the user and coach statement classifiers, including the size and label distribution in the train and test sets. The BERT-base model had 81% multi-class accuracy in accurately classifying the coach message as motivation, capability or opportunity.

To quantitatively evaluate the re-ranked response compared to the default response, 8 independent reviewers rated both the responses across several dimensions of activity coaching Table 8. Based on Likert scale responses across several dimensions, the re-ranked answers were rated better than unprimed [3.65±1.32 vs 3.04±0.95] with p-value confidence.

## Discussion

This proof-of-concept study introduces two methods to infuse behavior science into LLM dialogue. We demonstrate that behavior science-based priming is a simple but effective strategy to tailor LLMs for activity coaching, with specific benefits in terms of actionability and the provision of concrete and supporting coaching advice. Additionally, post-hoc re-ranking of LLM responses based on behavior science principles can further enhance attributes such as perceived empathy.

Coach phrase priming yielded significant boosts in various proxies for coaching quality. This trend was evident across both quantitative and qualitative metrics. Notably, coach phrase

**Table 7. Model performance on C/O/M classification for coach statements.**

| Category | Train | Test | Precision | Recall | F1 Score | Multi-class accuracy |
|---|---|---|---|---|---|---|
| Motivation | 256 | 121 | 0.87 | 0.86 | 0.86 | 0.81 |
| Capability | 139 | 66 | 0.88 | 0.72 | 0.79 | |
| Opportunity | 212 | 74 | 0.83 | 0.71 | 0.77 | |

**Table 8. Evaluation attribute ratings of coach-primed versus classifier re-ranked and oracle re-ranked dialogues.**

| Survey question (1, strong disagree → 5, strong agree) | Unprimed | Coach-primed | Classifier Re-ranked | Oracle Re-ranked | p value (re-ranked vs Unprimed) | Cohen-D (re-ranked vs Unprimed) | Beta (re-ranked vs Unprimed) |
|---|---|---|---|---|---|---|---|
| *response provided concrete fitness strategies that are actionable* | 2.88 ± 0.85 | 3.22 ± 0.95 | 3.66 ± 0.89 | 4.02 ± 0.68 | **0.01** | 0.63 | 0.78 |
| *response to user questions was in a realistic manner* | 3.02 ± 0.98 | 3.23 ± 0.97 | 3.59 ± 0.99 | 3.75 ± 0.80 | **0.01** | 0.37 | 0.5 |
| *response provided motivation or encouragement to the user* | 3.05 ± 1.0 | 3.19 ± 0.85 | 3.75 ± 0.92 | 3.45 ± 1.05 | **0.001** | 0.52 | 0.68 |
| *response is empathetic toward the user's needs and challenges* | 2.94 ± 0.99 | 3.05 ± 0.94 | 3.56 ± 0.87 | 3.77 ± 0.83 | **0.001** | 0.47 | 0.62 |
| *The language used is realistic and appropriate for the setting* | 3.33 ± 0.87 | 3.48 ± 0.84 | 3.69 ± 0.87 | 3.78 ± 0.79 | **0.014** | 0.29 | 0.36 |
| **Average total score** | 3.04 ± 0.95 | 3.24 ± 1.36 | 3.65 ± 1.32 | 3.75 ± 0.84 | | | |

priming was associated with a higher number of conversational turns, a greater rate of question-asking, and more frequent use of coaching vocabulary. Manual review also judged coach phrase priming as providing significantly greater motivation and concrete coaching strategies versus the unprimed LLM. This suggests that Coach phrase priming may be an effective and accessible strategy for customising LLMs for various coaching scenarios.

A unique aspect of this work is the combination of priming with post-hoc re-ranking to enable knowledge infusion at multiple touchpoints. Interestingly, re-ranking resulted in significant incremental improvements in actionability, with upward trends in empathy, motivation and realism that did not meet statistical significance. We demonstrate this uplift both for a classifier-based re-ranking, which can introduce error from mis-classification; and for oracle-based re-ranking, which showed a further marginal advantage over the former. Together, these results demonstrate the ability to stitch together multiple simple constraints as part of a hybrid knowledge infusion strategy. As LLMs become more pervasive in the coaching domain, this will be increasingly important.

Since Capability has marginally lower user statement classifier accuracy, it was wrongly identified as motivation in few cases of classifier based BeSci dialogue alignment LLM. This resulted in higher motivational character to classifier based LLM over Oracle LLM at the expense of lower empathy and actionability scores.

This study has a number of limitations. First, the evaluation was predominantly based on simulated conversations with a single human interacting with the LLMs, which invariably introduces bias even in the presence of blinding. Future work could trial a similar evaluation with larger groups of users engaging in the dialogue. The rudimentary priming method used here could be extended, e.g. by more explicit instruction prompting or chain of thought prompting. The re-ranking method was limited in only focusing on a single user query and coach response. In reality, it is important to consistently align the coach responses to user need throughout a conversation and adapt as the dialogue unfolds. Methods such as reinforcement learning with human feedback can help to offer this adaptability [15]. Finally, the behaviour model used was a simplistic one that conceptualizes user behaviour only along three axes—future studies could consider using more sophisticated behavior science frameworks, which may help to better target coach actions.

## Conclusion

Knowledge infusion methods based on behavior science principles can be used to improve the quality of LLM-generated physical activity related conversations. The combination of coach phrase priming with re-ranking of LLM outputs offers optimal results in terms of manually-adjudicated actionability, empathy and overall coaching experience. These methods can help to constrain and guide LLMs in various coaching scenarios.

## Supporting information

**S1 Table. BLEU match score to compare LLM primiring strategies to match human coach sentences.**
(PDF)

**S2 Table. User queries across COM themes selected for LLM evaluation.**
(PDF)

**S1 Text. Coach phrase priming sentence selection method.**
(PDF)

**S2 Text. LLM based conversation evaluation tool and methodology.**
(PDF)

**S3 Text. Coach response & user query classifier.**
(PDF)

**S4 Text. Summary of PACE study and adaptation to FIT-LLM work.**
(PDF)

**S1 Fig. LLM conversation rating tool used by annotators.**
(PDF)

**S2 Fig. Priming and BeSci infusion to LLM framework pipeline in Fit-LLM for user query input.**
(PDF)

## Acknowledgments

For technical and clinical advice and discussion, we thank the following, who are all employees of Alphabet Inc: Partha Talukdar, Hulya Emir Farinas, Cathy Speed, John Hernandez, Sriram Lakshminarasimhan. For software infrastructure, logistical support, and slide digitization services, we thank members of the Google Research and LaMDA teams. Lastly, we are deeply grateful to the annotation management team Rahul Singh and Ameena Khaleel.

## Author Contributions

**Conceptualization:** Narayan Hegde, Madhurima Vardhan, Emily Rosenzweig, Cathy Speed.

**Data curation:** Madhurima Vardhan.

**Formal analysis:** Madhurima Vardhan.

**Investigation:** Madhurima Vardhan, Deepak Nathani, Emily Rosenzweig.

**Methodology:** Narayan Hegde, Madhurima Vardhan, Deepak Nathani, Martin Seneviratne.

**Resources:** Narayan Hegde.

**Software:** Narayan Hegde.

**Supervision:** Narayan Hegde, Alan Karthikesalingam.

**Validation:** Emily Rosenzweig, Alan Karthikesalingam, Martin Seneviratne.

**Writing – original draft:** Narayan Hegde, Cathy Speed, Martin Seneviratne.

**Writing – review & editing:** Narayan Hegde, Emily Rosenzweig, Cathy Speed, Alan Karthikesalingam, Martin Seneviratne.

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
