## [Decision Letter · Decision Letter 0]

25 Aug 2023

PDIG-D-23-00123

Infusing behavior science into large language models for activity coaching

PLOS Digital Health

Dear Dr. Hegde,

Thank you for submitting your manuscript to PLOS Digital Health. After careful consideration, we feel that it has merit but does not fully meet PLOS Digital Health's publication criteria as it currently stands. Therefore, we invite you to submit a revised version of the manuscript that addresses the points raised during the review process.

Please submit your revised manuscript within 60 days Oct 24 2023 11:59PM. If you will need more time than this to complete your revisions, please reply to this message or contact the journal office at digitalhealth@plos.org. Please include the following items when submitting your revised manuscript:

We look forward to receiving your revised manuscript.

Kind regards,

Shlomo Berkovsky

Section Editor

PLOS Digital Health

Journal Requirements:

1. We ask that a manuscript source file is provided at Revision. Please upload your manuscript file as a .doc, .docx, .rtf or .tex.

Additional Editor Comments (if provided):

Reviewers' comments:

Reviewer's Responses to Questions

**Comments to the Author**

1. Does this manuscript meet PLOS Digital Health’s publication criteria? Is the manuscript technically sound, and do the data support the conclusions? The manuscript must describe methodologically and ethically rigorous research with conclusions that are appropriately drawn based on the data presented.

Reviewer #1: Yes

Reviewer #2: Yes

2. Has the statistical analysis been performed appropriately and rigorously?

Reviewer #1: Yes

Reviewer #2: Yes

3. Have the authors made all data underlying the findings in their manuscript fully available (please refer to the Data Availability Statement at the start of the manuscript PDF file)?

Reviewer #1: No

Reviewer #2: No

4. Is the manuscript presented in an intelligible fashion and written in standard English?

Reviewer #1: Yes

Reviewer #2: No

5. Review Comments to the Author

Reviewer #1: The manuscript titled 'Infusing behavior science into large language models for activity coaching' was an interesting read, delving into a topic that seemed quite interesting and relevant. The authors explored how ideas from behavioral science may be used to enhance LLM conversations in the physical activity domain. The study is well constructed and fairly explained. However, I have some concerns which I would like the authors to modify/clarify.

1. More context to the PACE study. The authors introduce the PACE study in the introduction abruptly, without sufficient context. The explanation given on what the original study was designed for seemed a little too brief. More context to the dataset and how it was used previously would be useful.

2. Many abbreviations are laid out without explanation (BERT, FBM, PACE). While some may be standard, it would still be important to mention what they mean.

3. I wonder if 'qualitative review' is appropriate for the review using Likert scales. I understand what the authors mean, but I would still suggest alternative phrasing for a review that includes numerical data.

4. The authors mention a series of linear mixed model analyses. While the model is described briefly, I think a model equation laying out the fixed and random effects might make this much clearer. Moreover, I was not sure where the results of the model are, i.e the coefficients of each of the predictors.

5. Post-hoc re-ranking did not seem to show significant difference from the coach primed dialogues in multiple ranking criteria. The authors could perhaps elaborate why they believe this happened/what this implies (insufficient power/why certain criteria are less likely to benefit from re-ranking).

6. While p-values are useful, effect sizes might indicate how relevant the differences are. This may be more important when values are marginally significant/insignificant. It would be nice if the authors could add values for the effect size, quantified using say, the Cohen's d, for the t-tests.

Minor edits

There seem to be quite a few minor issues scattered throughout the manuscript. A few glaring ones are noted below

1. Consistency in referring to Supplementary material, Tables etc. 

2. Page 3, para 1. "consented subjects...". Consenting may be more appropriate.

3. Page 3, para 1. "consented subjects were randomized to coaches or coaches...". coaches or coaches? Rephrase for clarity

4. Page 7, para 2. A line reads simply "The (Table 5)."

5. Page 8, para 5. "Future work could trial a similar evaluation with larger groups of users engaging in

dialogue, as per [ref].", missing reference.

Reviewer #2: Summary: The goal of the authors was to integrate the COM-B principle of behavioral science into LLM through two infusion techniques: coach message priming and dialogue re-ranking. The aim was to assist users in adopting a more active lifestyle by using behavioral nudges and conversational solutions to overcome barriers.

Comments:

1. I appreciate the Authors’ efforts on this paper.

2. Authors need to justify why defining the evaluation metrics is a contribution.

3. In this paper, the authors incorporated principles and techniques from two disciplines, behavior science and computer science that serve others in other field like digital health. Integrating two disciplines is a great idea, but the reader, a specialist in one discipline, is mostly unfamiliar with the other. More shedding light on the definitions and explanations of the terms can be more beneficial to readers from different disciplines, and adding a background section might be helpful. The terms are large language model, task-oriented dialogue, behavior science framework, priming, primed LLM, and unprimed LLM.

4. Adding a literature review section will show how the work is novel and different from the others.

5. As mentioned in the introduction, you extend the work of PACE. Give a summary of the PACE study.

6. Providing a screenshot or an example of one dataset record will give the reader some visualization of the dataset, especially the availability of data is restricted to some researchers.

7. Wizard of Oz protocol needs to be defined and give background about it.

8. Some acronyms like FBM and BS need the whole sentence.

9. Give more explanation about LaMDA.

10. Is there any reason for keeping the model in its primary architecture without further tuning?

11. What interface is used for the proposed approach? Text, speech, or multi-modal?

12. More elaboration about the coach message priming and dialogue ranking is needed.

13. Text in Figure 1 needs to be clearer.

14. Why did you use the cross entropy optimizer, not other?

6. PLOS authors have the option to publish the peer review history of their article (what does this mean?). If published, this will include your full peer review and any attached files.

**Do you want your identity to be public for this peer review?** For information about this choice, including consent withdrawal, please see our Privacy Policy.

Reviewer #1: No

Reviewer #2: No

---

## [Decision Letter · Decision Letter 1]

3 Nov 2023

PDIG-D-23-00123R1

Infusing behavior science into large language models for activity coaching

PLOS Digital Health

Dear Dr. Hegde,

Thank you for submitting your manuscript to PLOS Digital Health. After careful consideration, we feel that it has merit but does not fully meet PLOS Digital Health's publication criteria as it currently stands. Therefore, we invite you to submit a revised version of the manuscript that addresses the points raised during the review process.

Please submit your revised manuscript within 30 days Dec 03 2023 11:59PM. If you will need more time than this to complete your revisions, please reply to this message or contact the journal office at digitalhealth@plos.org. Please include the following items when submitting your revised manuscript:

We look forward to receiving your revised manuscript.

Kind regards,

Crina Grosan

Academic Editor

PLOS Digital Health

Journal Requirements:

2. Please ensure that Funding Information and Financial Disclosure Statement are matched.

3. In the Funding Information you indicated that no funding was received. Please revise the Funding Information field to reflect funding received.

4. Please provide separate figure files in .tif or .eps format only and remove any figures embedded in your manuscript file. Please also ensure that all files are under our size limit of 10MB.

Additional Editor Comments (if provided):

Reviewers' comments:

Reviewer's Responses to Questions

**Comments to the Author**

1. If the authors have adequately addressed your comments raised in a previous round of review and you feel that this manuscript is now acceptable for publication, you may indicate that here to bypass the “Comments to the Author” section, enter your conflict of interest statement in the “Confidential to Editor” section, and submit your "Accept" recommendation.

Reviewer #1: All comments have been addressed

Reviewer #2: (No Response)

2. Does this manuscript meet PLOS Digital Health’s publication criteria? Is the manuscript technically sound, and do the data support the conclusions? The manuscript must describe methodologically and ethically rigorous research with conclusions that are appropriately drawn based on the data presented.

Reviewer #1: Yes

Reviewer #2: (No Response)

3. Has the statistical analysis been performed appropriately and rigorously?

Reviewer #1: Yes

Reviewer #2: Yes

4. Have the authors made all data underlying the findings in their manuscript fully available (please refer to the Data Availability Statement at the start of the manuscript PDF file)?

Reviewer #1: (No Response)

Reviewer #2: Yes

5. Is the manuscript presented in an intelligible fashion and written in standard English?

Reviewer #1: Yes

Reviewer #2: Yes

6. Review Comments to the Author

Reviewer #1: The authors have now addressed most of the concerns I raised in the previous version. One minor addition, in section 2.8 the authors now give the equation for the models, but no explanation for what the coefficients are. I'd suggest that the authors do this during the proof stage or next submission.

Reviewer #2: Thanks to the authors for their efforts in addressing most of the comments.

The paper needs to provide a literature review section showing how your work is novel and distinctive from other studies in the same field.

7. PLOS authors have the option to publish the peer review history of their article (what does this mean?). If published, this will include your full peer review and any attached files.

**Do you want your identity to be public for this peer review?** For information about this choice, including consent withdrawal, please see our Privacy Policy. 

Reviewer #1: Yes: Sandip V George

Reviewer #2: No

---

## [Decision Letter · Decision Letter 2]

14 Dec 2023

Infusing behavior science into large language models for activity coaching

PDIG-D-23-00123R2

Dear Mr Hegde,

We are pleased to inform you that your manuscript 'Infusing behavior science into large language models for activity coaching' has been provisionally accepted for publication in PLOS Digital Health.

Best regards,

Crina Grosan

Academic Editor

PLOS Digital Health

Reviewer Comments (if any, and for reference):

Reviewer's Responses to Questions

**Comments to the Author**

1. If the authors have adequately addressed your comments raised in a previous round of review and you feel that this manuscript is now acceptable for publication, you may indicate that here to bypass the “Comments to the Author” section, enter your conflict of interest statement in the “Confidential to Editor” section, and submit your "Accept" recommendation.

Reviewer #2: All comments have been addressed

2. Does this manuscript meet PLOS Digital Health’s publication criteria? Is the manuscript technically sound, and do the data support the conclusions? The manuscript must describe methodologically and ethically rigorous research with conclusions that are appropriately drawn based on the data presented.

Reviewer #2: Yes

3. Has the statistical analysis been performed appropriately and rigorously?

Reviewer #2: I don't know

4. Have the authors made all data underlying the findings in their manuscript fully available (please refer to the Data Availability Statement at the start of the manuscript PDF file)?

Reviewer #2: Yes

5. Is the manuscript presented in an intelligible fashion and written in standard English?

Reviewer #2: Yes

6. Review Comments to the Author

Reviewer #2: Thanks to the authors for their efforts in addressing all the comments.

7. PLOS authors have the option to publish the peer review history of their article (what does this mean?). If published, this will include your full peer review and any attached files.

**Do you want your identity to be public for this peer review?** For information about this choice, including consent withdrawal, please see our Privacy Policy.

Reviewer #2: No
